# A Mucoralean White Collar-1 Photoreceptor Controls Virulence by Regulating an Intricate Gene Network during Host Interactions

**DOI:** 10.3390/microorganisms9020459

**Published:** 2021-02-23

**Authors:** Carlos Pérez-Arques, María Isabel Navarro-Mendoza, Laura Murcia, Carlos Lax, Marta Sanchis, Javier Capilla, Eusebio Navarro, Victoriano Garre, Francisco Esteban Nicolás

**Affiliations:** 1Departmento de Genética y Microbiología, Facultad de Biología, Universidad de Murcia, 30100 Murcia, Spain; carlos.perez6@um.es (C.P.-A.); mariaisabel.navarro3@um.es (M.I.N.-M.); lauramur@um.es (L.M.); carlos.lax@um.es (C.L.); sebi@um.es (E.N.); 2Unidad de Microbiología, Universitat Rovira i Virgili, IISPV, 43003 Tarragona, Spain; marta.sanchis@urv.cat (M.S.); javier.capilla@urv.cat (J.C.)

**Keywords:** white collar, mucormycosis, virulence, Mucorales, *Mucor lusitanicus*

## Abstract

Mucolares are an ancient group of fungi encompassing the causal agents for the lethal infection mucormycosis. The high lethality rates, the emerging character of this disease, and the broad antifungal resistance of its causal agents are mucormycosis features that are alarming clinicians and researchers. Thus, the research field around mucormycosis is currently focused on finding specific weaknesses and targets in Mucorales for developing new treatments. In this work, we tested the role of the *white-collar* genes family in the virulence potential of *Mucor lusitanicus*. Study of the three genes of this family, *mcwc-1a*, *mcwc-1b*, and *mcwc-1c*, resulted in a marked functional specialization, as only *mcwc-1a* was essential to maintain the virulence potential of *M. lusitanicus*. The traditional role of *wc-1* genes regulating light-dependent responses is a thoroughly studied field, whereas their role in virulence remains uncharacterized. In this work, we investigated the mechanism involving *mcwc-1a* in virulence from an integrated transcriptomic and functional approach during the host–pathogen interaction. Our results revealed *mcwc-1a* as a master regulator controlling an extensive gene network. Further dissection of this gene network clustering its components by type of regulation and functional criteria disclosed a multifunctional mechanism depending on diverse pathways. In the absence of phagocytic cells, *mcwc-1a* controlled pathways related to cell motility and the cytoskeleton that could be associated with the essential tropism during tissue invasion. After phagocytosis, several oxidative response pathways dependent on *mcwc-1a* were activated during the germination of *M. lusitanicus* spores inside phagocytic cells, which is the first stage of the infection. The third relevant group of genes involved in virulence and regulated by *mcwc-1a* belonged to the “unknown function,” indicating that new and hidden pathways are involved in virulence. The unknown function category is especially pertinent in the study of mucormycosis, as it is highly enriched in specific fungal genes that represent the most promising targets for developing new antifungal compounds. These results unveil a complex multifunctional mechanism used by *wc-1* genes to regulate the pathogenic potential in Mucorales that could also apply to other fungal pathogens.

## 1. Introduction

Mucormycosis is a threatening infection caused by Mucorales that is the most unexplored host–pathogen interaction [1]. The recent interest in mucormycosis relies on the three main features of this infection: high mortality rates, its emerging character, and its unusual antifungal resistance. The direct cause of the high mortality rates is the lack of effective antifungal compounds against mucormycosis [2]. The fungi of the order Mucorales are resistant to most of the current antifungal compounds [3,4]. The high mortality rates, the increasing incidence, and the recent antifungal resistances found in Mucorales reveal the necessity of new and more effective treatments [5]. In this context, most of the current research is based on searching for and characterizing new targets in the physiology of Mucorales that could serve for the development of new antifungal compounds. Different research lines study the role of virulence factors described in Mucorales pathogenesis, such as the CotH proteins, the three components of the high-affinity iron uptake system, and the regulatory proteins involved in dimorphism [6,7,8,9,10]. In addition, several genome-wide strategies have been designed to find new and specific targets in Mucorales, based on the RNAi mechanism of the fungus [11] and in the host–pathogen transcriptome [12,13]. This work aims to study the role of the *white collar-1* (*wc-1*) genes in the virulence of Mucorales as master regulators controlling many pathways that could represent new promising targets. The family of *wc-1* genes has been associated with virulence in both Ascomycetes and Basidiomycetes [14,15,16]; however, it never has been studied in Mucorales.

The *wc-1* genes control most of the physiological responses of fungi to light. Mutations in these genes provoke the lack of light-regulated responses like the production of carotenes, the control of the circadian clock, sporulation, and phototropism [17,18,19]. The structure of the proteins encoded by these genes possesses two specific domains: Per-Arnt-Sim (PAS) domains and a zinc-finger DNA-binding domain. One of its three PAS domains specializes in a light, oxygen, or voltage (LOV) domain. The LOV domain is a light receptor, and the zinc-finger domain recognizes the elements responding to light found in the promoters of target genes [20]. Thus, the White Collar-1 proteins are both DNA binding transcription factors and light sensors [21]. The role of the *wc-1* genes regulating different responses to light has been intensely studied and dissected in several fungi since they were firstly described in *Neurospora crassa* [22]. However, their role in fungal virulence is a recent discovery described in fungi such as *Fusarium oxysporum*, *Cryptococcus neoformans*, *Cercospora zeae-maydisa*, and *Botrytis cinerea* [14,15,16,23]. These studies showed that the role of the *wc-1* genes in virulence is the same in different fungi, with these genes being necessary to maintain full virulence. In addition, *wc-1* genes regulate virulence in both plant- and animal-specific fungal pathogens. In *F. oxysporum*, which infects both plant and animal hosts, lack of *wc-1* gene only affected virulence in animal hosts [16]. Moreover, studies made in *C. neoformans* have suggested that the role of *wc-1* genes in virulence is independent of light [14]. In these studies, a mutational analysis of the different domains in *wc-1* proteins showed that light-sensing impaired mutants were still fully virulent [14].

In Mucorales, there are no studies relating the *wc-1* genes to virulence. However, several studies of the fungus *Mucor lusitanicus* showed a detailed dissection of a complex *wc-1* genes regulatory system, characterizing all the responses controlled by light and these genes [24]. In *M. lusitanicus*, there are three different orthologous genes, which acquired functional specialization concerning the different responses triggered by light [24]. These three genes were denominated *mcwc-1a*, *mcwc-1b*, and *mcwc-1c*. First, mutagenesis analysis of these three genes showed that mutants in *mcwc-1c* were defective in the carotene production, whereas *mcwc-1a* mutants lacked the phototropism response [24]. A second study found that *mcwc-1b* is involved in regulating carotene production and vegetative growth, but in a more complex pathway, including the negative regulator gene *crgA* [25,26,27,28].

Here, we explored the role of the *wc-1* genes in the virulence of Mucorales, finding that indeed virulence is also dependent on this family of genes. As expected from the functional specialization of the three *mcwc-1* genes in the responses triggered by light, virulence is specifically controlled by only one of the three genes, *mcwc-1a*. After considering these results, we analyzed the pathways regulated by *wc-1* genes related to virulence, which have not been explored in any other fungi. Our group previously developed a system to study host–pathogen interactions using fungal spores phagocyted by macrophages [11]. In this system, we investigated the transcriptional response dependent on *mcwc-1a* during phagocytosis, finding an intricate gene network involved in the process.

## 2. Materials and Methods

### 2.1. Fungal Strains and Cell Cultures

*M. lusitanicus* strain CBS277.49 served as the parental strain of all the following strains. The mutants in *white collar-1* genes (*mcwc-1a*∆ (MU242), *mcwc-1b*∆ (MU244), and *mcwc-1c*∆ (MU247)) were generated in a previous study [24], derived from the leucine and uracil auxotrophic strain MU402, by rescuing the uracil auxotrophy. MU402 was derived by mutagenesis from the leucine auxotrophic strain R7B, which was used as the wild-type control strain for RNA-seq analyses and virulence assays [29] because it harbors the same genetic background and leucine auxotrophy as the *wc-1* mutants, except for the aforementioned deletions. *M. lusitanicus* strain NRRL3631 is an independent natural isolate that was used as an avirulent mock control in the virulence assays [30].

Fungal colonies were grown in a solid, rich YPG medium with pH adjusted at 4.5 and an incubation temperature of 26 °C for optimal growth and sporulation [31]. Spores were harvested after six days and then filtered using Falcon^®^ 70 µm cell strainers to remove unintentionally harvested mycelium pieces before confronting macrophages or animal models.

### 2.2. Virulence Assays

Male OF-1 mice weighing 30 g and two months old (Charles River, Barcelona, Spain) [30,32,33] were used as animal models for virulence assays. The mice were immunosuppressed by intraperitoneal administration of cyclophosphamide (200 mg/kg of body weight), 2 days prior to infection and then once every 5 days. Eight-mouse groups were challenged intravenously by retroorbital injection of 1 × 10^6^ spores [32]. The assay was repeated twice using two independent mutants of *mcwc-1a* (MU242 and MU243). During this procedure, the mice were anesthetized using isoflurane via inhalation and monitored until they recovered from the anesthesia. Mice were housed under established conditions with free food and autoclaved water. The animal welfare was checked twice a day for 20 days, and those mice following the criteria for discomfort were euthanized by CO_2_ inhalation. Survival rates during the time were plotted in a Kaplan–Meier curve (Graph Pad Prism), and differences were considered statistically significant with a *p*-value ≤ 0.05 in a Mantel–Cox test.

### 2.3. Host–Pathogen In Vitro Assays, RNA Purification, and Sequencing

Spores from mutant strain *mcwc-1a*∆ and the wild type R7B were confronted with a mouse macrophage cell-line (J774A.1 or ATCC TIB-67) in a 1.5:1 ratio as described in previous studies to perform host–pathogen interaction assays [12]. Briefly, the confrontation occurred in L15 medium supplemented with 20% fetal bovine serum (both from Capricorn Scientific) at a temperature of 37 °C for 5 h to ensure all spores were phagocytosed. As a non-confrontation control or saprotrophic conditions, the same concentration of spores was cultured in L15 medium supplemented with fetal bovine serum. Samples were prepared in quadruplicate.

Duplicates of each sample were pooled together, and RNA was extracted using the standard procedure of the RNeasy plant minikit (Qiagen, Hilden, Germany), resulting in two replicated RNA extractions per sample that were sent for sequencing to BaseClear (Leiden, The Netherlands). There, mRNA was enriched by poly(A) purification capture, and cDNA libraries were prepared using TruSeq RNA library preparation kits. The cDNA library was sequenced using an Illumina HiSeq 2500 system, producing two replicated raw datasets of first-stranded (or reverse), 50 bp, single-end reads.

### 2.4. RNA-Sequencing Analysis for Gene Expression

The quality of the raw reads from *M. lusitanicus* wild-type and *mcwc-1a*∆ single- and co-cultures with mouse macrophages was assessed with FASTQC v0.11.8 before and after removing adapter and contaminant sequences using Trim Galore! v.0.6.2 (available at http://www.bioinformatics.babraham.ac.uk/projects/ (accessed on 19 February 2021)). Reads with low Phred quality score (Q ≤ 32) and/or total length (length ≤ 20 nt) were filtered out, as well as adapter sequences with an overlap longer than ≥4 bases in the 3′ end of the read. These high-quality and adapter-free reads were mapped to the *M. lusitanicus* f. *lusitanicus* v2.0 genome (herein Mucci2 [33], available at https://mycocosm.jgi.doe.gov/Mucci2/Mucci2.info.html (accessed on 19 February 2021)) using STAR v.2.7.1a [34]. Read counts were calculated by HTSeq v.0.9.1 [35] excluding multi-mapping reads and were used to perform a differential gene expression analysis with the *limma* package v.3.38.3 [36], using robust settings for empirical Bayes statistics and the trimmed mean of M values (TMM) method for sample normalization. Genes were considered differentially expressed (DEGs) when meeting the following criteria used in previous studies [12]: false discovery rate (FDR) ≤ 0.05, log2 fold change (log_2_ FC) ≥ 1.0, and average count per million reads (CPM) in log2 units ≥ 0.0). Average gene expression values (as log_2_ CPM) and log_2_ FC ratios were correlated among samples with scatter plots, highlighting up- and down-regulated genes and three housekeeping genes whose expression is equal among samples: translation elongation factor EF-1 (Mucci2 ID: 156959), RNA polymerase III transcription factor IIIC subunit TFIIIC (Mucci2 ID: 106349), and vacuolar-type H^+^-ATPase V-ATPase (Mucci2 ID: 154376).

DEGs were clustered with Canberra distance and Ward’s clustering method to plot a heatmap using the pheatmap package v1.0.12 (available at https://cran.r-project.org/web/packages/pheatmap/ (accessed on 19 February 2021)). Eukaryotic Orthologous Groups (KOG) and Gene Ontology (GO) terms were identified with EggNOG-mapper v2.0 [37,38]. This classification was used to conduct a KOG class enrichment analysis using KOGMWU package v1.2 [39]; the difference between the mean rank of all genes within the KOG class and the mean rank of all other genes (or delta-rank) was calculated with a Mann–Whitney U-test to estimate up- or down-regulation of any KOG class. Then, DEGs were compared to the total amount of genes in each KOG class, considering a significant enrichment in that particular KOG class if *p*-value ≤ 0.05 in a one-sided Fisher’s exact test. GO term enrichment was also analyzed using FungiDB (available at https://fungidb.org/fungidb/app (accessed on 19 February 2021)), plotting the enriched and non-redundant GO terms with REVIGO (available at http://revigo.irb.hr/ (accessed on 19 February 2021)).

## 3. Results

### 3.1. The Gene Mcwc-1a Is Essential to Maintain the Virulence Potential in M. lusitanicus

The *mcwc-1a*Δ, *mcwc-1b*Δ, and *mcwc-1c*Δ mutants were tested in a survival assay using an immunosuppressed mouse model of infection. Mice were also infected with two wild-type control strains: the virulent R7B strain and the avirulent NRRL3631 strain, as control strains. The results of these survival assays are presented in Figure 1. The lack of the gene *mcwc-1a* resulted in a significant reduction in the virulence potential in *M. lusitanicus* (Mantel–Cox test, *p*-value = 0.006). In contrast, lacking any of the other two *wc-1* genes, *mcwc-1b* or *mcwc-1c*, did not alter the virulence potential of *M. lusitanicus* because the corresponding mutants showed a similar pathogenic behavior as the virulent wild-type R7B strain (survival percentages: *mcwc-1a*Δ 75%, *mcwc-1b*Δ 25%, *mcwc-1c*Δ 25%, R7B 12.5% and NRRL3631 100%). These results indicated that in the *M. lusitanicus wc-1* gene family, the gene *mcwc-1a* has a specialized role in virulence, although a partial redundancy of the other two genes cannot be excluded.

### 3.2. A High-Throughput Transcriptomic Analysis Reveals the Gene Network Controlled by mcwc-1a

After identifying the essential role of the gene *mcwc-1a* in the virulence of Mucorales and its consequent functional conservation throughout the fungal kingdom, we explored the transcriptional response controlled by *mcwc-1a* both during regular growth and macrophage phagocytosis, trying to identify the genes required for the fungal response during the first stage of the host–pathogen interaction. We conducted a deep-sequencing approach (RNA-seq) using mRNA from spores of the wild-type R7B strain and the mutant *mcwc-1a*Δ grown in two different conditions (both in the dark). The first condition mimicked a saprotrophic environment, using the rich medium L15. The second condition was a co-culture of spores and the J774A.1 cell-line of mouse macrophages (1.5:1 spore–macrophage ratio) for 5 h, ensuring the phagocytosis of all the spores. This second condition is a novel study model in Mucorales to investigate the initial step of fungal infections, in which the pathogen must overcome the innate immune response to continuing the infection and tissue invasion [1]. The scatter plots of gene expression values in the two conditions described above (Figure 2A,B) reveal that the *mcwc-1a*Δ mutant had significantly different profiles in the two growth conditions. The analysis showed that a reduced proportion of the genes controlled by *mcwc-1a* are involved in response to phagocytosis, the first step of the fungal infection (Table 1). We selected differentially expressed genes (DEGs) as those surpassing a corrected *p*-value of less than or equal to 0.05 (false discovery rate (FDR) ≤ 0.05) and an absolute log_2_ fold change (FC) greater than or equal to 1 (log_2_ FC ≥ |1.0|). Thus, a total of 3456 genes (Appendix A) were directly or indirectly regulated by *mcwc-1a* (1957 upregulated and 1499 downregulated) during saprotrophic conditions. In contrast, after phagocytosis, only 478 genes showed differential expression controlled by this gene (172 upregulated and 306 downregulated) (Table 1). There are 11,719 genes annotated, meaning that these genes correspond to 29% during saprotrophic conditions (17% upregulated and 12% downregulated) and 4% in control conditions (1% upregulated and 3% downregulated).

### 3.3. Cellular Processes Regulated by mcwc-1a

Once the genes regulated by *mcwc-1a* were identified, we performed an enrichment analysis of Eukaryotic Orthologous Groups (KOG), exploring the putative cellular processes controlled by this gene both under saprotrophic conditions (rich medium) and macrophage phagocytosis (Figure 3). This analysis was intended to unveil possible cellular processes that could be involved in virulence and controlled by a transcription factor traditionally related to processes triggered by light. Under saprotrophic conditions without macrophages, several cellular processes showed a significant enrichment. Thus, processes related to the cytoskeleton and cell motility, metabolism and transport of coenzymes, and most importantly, signal transduction are significantly downregulated, probably due to the lack of Mcwc-1a activity. Furthermore, the metabolism of macromolecules (amino acids, carbohydrates, lipids), energy (oxidation-reduction), and cell wall and membrane biogenesis are altered (most of them showing upregulation). Under conditions with macrophages, the lower number of genes identified probably precludes the detection of cellular processes showing enrichment. Nevertheless, two cellular processes revealed statistical significance. The first group of genes is classified in the unknown function category, an especially relevant feature of Mucorales, as this group is unusually large compared with other fungi, and most of its components are specific to Mucorales [33]. The second category contains genes involved in metabolism and inorganic ion transport, which may protect from oxidative damage.

### 3.4. In-Depth Analysis of Genes Regulated by mcwc-1a during the Interaction with the Host

To further delimitate the function of those genes regulated during the interaction with macrophages, we analyzed the distribution of genes after the different regulation types exerted by *mcwc-1a* (Figure 4). After grouping genes under similar regulatory categories, i.e., sharing similar relative expression values, the study was followed by a GO term enrichment analysis (Figure 5). Only the genes differentially expressed after phagocytosis are shown here, and we have classified them into four clusters (Appendix A contains all genes; Appendix A, cluster 1; Appendix A, cluster 2; Appendix A, cluster 3; and Appendix A, cluster 4). The first one (Figure 4, purple-colored cluster) showed a group of 39 genes that could be directly related to pathogenic processes. It comprises those that are induced in the wild-type virulent strain with macrophages but are downregulated in the *mcwc-1a* mutant with macrophages. Those genes are most likely primary targets of Mcwc-1a and could be directly involved in the virulent response to macrophages. The analysis of GO terms (Figure 5) showed enrichment in antioxidant activity, acting on a group of sulfur donors (which would be related to redox reactions and protection against oxidative damage). Besides, we also found enrichment for signaling receptor activity and molecular transducer activity, which is in congruence with the transcriptional factor activity of WC1a.

The second cluster is the most numerous, counting 267 genes (Figure 4, green-colored cluster). In the wild-type strain, these genes are repressed during the interaction with macrophages and highly downregulated in the *mcwc-1a* mutant. In the GO term analysis, they are related to iron ion transmembrane transport, which is essential for virulence in Mucorales [7,40,41]. In addition, other cellular processes were represented, such as GTPases signaling, phosphotransferase activity, cholesterol delta-isomerase activity, and protein-dimerization activity (Figure 5).

The third cluster contains 118 genes induced in the wild-type strain when phagocytosed but shows further induction in the *mcwc-1a* mutant in the same conditions (Figure 4, yellow-colored cluster). This cluster is enriched in genes involved in several categories of metabolism, such as acetyl-coA, carboxylase activity, and transporter activity (Figure 5).

The fourth cluster contains 54 genes showing analogous expression to the third cluster, similarly induced in the wild-type and mutant strains in saprophytic conditions, but in this case, a further increase in the mutant during phagocytosis is observed (Figure 4, red color cluster). In this case, the metabolism of porphyrins (porphobilinogen synthase) stands out in the GO term analysis (Figure 5). Porphyrins are required for the heme groups, which are necessary for redox reactions; therefore, they play an essential role in oxidative damage protection [42].

## 4. Discussion

We analyzed the role of the three mc*wc-1* genes during host interactions and showed that *mcwc-1a* is involved in the virulence potential of *M. lusitanicus*. Previous works demonstrated the role in virulence of *wc-1* genes in different fungi of Basidiomycota and Ascomycota divisions [14,15,16]. Our results added the analysis of the *wc-1* genes to the evolutionary tree of pathogenic fungi, showing its highly conserved function from the early-diverging fungi Mucorales to the more recent Basidiomycetes. Besides the high conservation, the essential role of *wc-1* genes in both plant and animal fungal pathogens has evidenced the fundamental functions of these genes in the transition from saprophytic to pathogenic fungi, independently of the host type. In the specific case of the exceptional fungus *F. oxysporum*, which infects both plants and animals, the *wc-1* gene controls virulence only in animal hosts, suggesting that different genes and pathways assumed the virulent role in plant hosts [16].

The *wc-1* genes are a photoreceptor family that controls most of the physiological changes triggered in the mycelium after the light exposition. In a saprophytic environment, like a soil rich in organic material, the mycelia that reach the surfaces encounter the stressing conditions of sunlight and air exposition. The rapid response controlled by the *wc-1* genes activates the production of secondary metabolites like carotenes as protection for the UV light damage and the production of asexual spores for air spreading over a large area. In this context, it is difficult to relate the pathways and processes controlled by WC proteins with virulence regulation. However, the functional dissection of the WC structural domains has elucidated the evolution of the WC proteins that has resulted in virulence regulation. Two independent studies demonstrated in two different fungal models, *C. neoformans* and *Fusarium asiaticum*, that the photoreception domains are not required in virulence, whereas the DNA binding zinc-finger domains are essential for full virulence [14,43]. These results could explain why the *mcwc-1b* gene of *M. lusitanicus* is not essential in virulence because this gene encodes for a non-canonical zinc-finger domain that might be involved only in the regulation of carotenogenesis [40]. The three *wc-1* genes of *M. lusitanicus*, *mcwc-1a*, *mcwc-1b*, and *mcwc-1c*, have acquired a specific functional specialization. Thus, *mcwc-1a* regulates phototropism and virulence, whereas *mcwc-1b* and *mcwc-1c* control carotenogenesis by ubiquitination or light induction, respectively [25,26]. The existence of gene families with different paralogues exhibiting a functional specialization has been described several times in *M. lusitanicus* [7,41,44,45,46]. A study in the fungus *F. asiaticum* reported a similar functionalization of the *wc-1* genes, showing that only one of its two WC proteins is involved in virulence [16].

Once the conservation of the essential role of WC proteins in virulence has been established in Mucorales, the next question to resolve relates to the mechanism used by these proteins to regulate virulence. In *F. oxysporum*, a hypothesis based on the production of secondary metabolites under dark conditions was suggested as a possible explanation for the lack of virulence in the *wc-1* mutant [16]. Other studies showed that the production of gibberellins and bikaverins required a WC-1 protein in a light-independent manner in the fungus *Fusarium fujikuroi*, but without a clear link to virulence [47]. In *M. lusitanicus*, the genes *mcwc-1b* and *mcwc-1c* regulate the production of carotenes (which are the main secondary metabolites) without affecting virulence. However, these results do not discard the regulation of *mcwc-1a* of other unknown secondary metabolites with a putative role in virulence. Our results showing the functional specialization of the three *wc-1* genes in *M. lusitanicus* could suggest another hypothesis for its role in virulence. The gene related to virulence, *mcwc-1a*, also controls phototropism. Tropism is an essential process for growth and sporulation and also could be related to the pathogenic potential because it might control the hyphae movement during tissue and blood vessel invasion for dissemination. Thus, a set of genes involved in tropism and controlled by *mcwc-1a* could link to virulence.

Independently of how the WC-dependent mechanism works in regulating virulence, all the studies conclude that finding the WC-dependent genes is the necessary next step to understand this mechanism. Our results investigating the gene network controlled by *mcwc-1a* showed a substantial number of genes (almost one-third of the total predicted genes) differentially expressed during saprotrophic growth conditions. The general interpretation of these results clearly defines the *wc-1* genes as master regulators controlling the development of *M. lusitanicus*. However, a more specific and reduced set of genes was found differentially expressed during the interaction with macrophages, in a context simulating the first stage of the infection. A detailed analysis clustering all the differentially expressed genes and dissecting their putative functional associations led us to suggest three non-exclusive hypotheses to explain the role of WC proteins in controlling virulence. First, we have repeatedly found groups of genes in different clusters involved in the protection against oxidative damage. This process is an essential fungal function required both after the exposition to the light and during the macrophage confrontation, linking the traditional response of WCs with its role in controlling virulence. Thus, our results suggest that *mcwc-1a* might be influencing the oxidative-damage protection pathways in the two scenarios. Secondly, we found an enrichment of differentially expressed genes with unknown functions during the interaction with the macrophages. These results indicate that besides the protection against oxidative stress, *mcwc-1a* must be controlling other cellular processes with essential roles in the survival inside the macrophages. This group of unknown function genes in Mucorales contains a large proportion of unique genes not found in other organisms [33], which makes them the perfect candidates to find targets for new antifungal compounds. Finally, our results show an important enrichment of genes involved in cell motility and the cytoskeleton out of the interaction with macrophages. These results could suggest a role of *mcwc-1a* in the second stage of the infection, during the tissue invasion, angioinvasion, and dissemination of the infection. This stage could link to essential processes related to tropism, cell motility, and cytoskeleton-reorganization functions. These could be the first results suggesting a link between the regulation of phototropism and virulence by Mcwc-1a in a light-independent manner.

The virulence potential of fungal pathogens depends on complex mechanisms controlled by diverse pathways and cellular processes. The intricate gene network controlled by *mcwc-1a* described in this work highlights the oxidative stress, tropism, and a group of unknown functions as the main processes that this master regulator might control to regulate virulence potential in fungal pathogens.

## Figures and Tables

**Figure 1 microorganisms-09-00459-f001:**
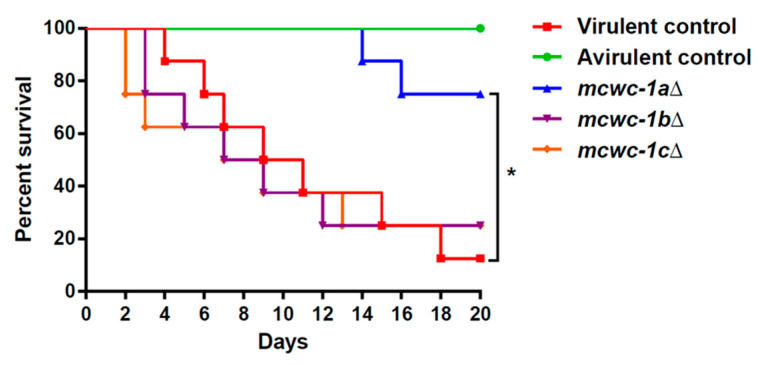
The gene *mcwc-1a* is essential to maintain the virulence potential. Virulence assays in immunosuppressed mice injected with 1 × 10^6^ spores of the mutants in the three *wc-1* genes of *M. lusitanicus*: *mcwc*-1aΔ (blue), *mcwc*-1bΔ (purple), and *mcwc*-1cΔ (orange). In addition, two control strains were also injected: the virulent wild-type strain R7B (red) and the avirulent wild-type strain NRRL3631 (green). The survival rate of the mutant was compared to the virulent control strain and statistically analyzed by a Mantel–Cox test (*, *p*-value = 0.006).

**Figure 2 microorganisms-09-00459-f002:**
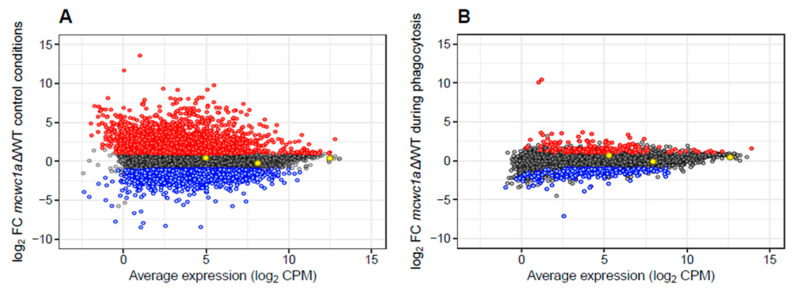
Scatter plots of gene expression values in the *mcwc-1a*Δ mutant at 5 h during regular growth (**A**) and macrophage phagocytosis (**B**) compared to gene expression in the wild-type strain. Each dot represents the expression value of any given gene as CPM (in log_2_ units, x-axis) and the fold change (FC) compared to the wild-type expression value (in log_2_ units, y-axis). Genes color-coded as red or blue indicate upregulated or downregulated significant genes (FDR ≤ 0.05, average log_2_ CPM > 0, log_2_ FC ≥ |1|), respectively. Three housekeeping genes—encoding EF-1, TFIIIC, and V-ATPase—are shown colored as yellow to assure proper normalization among samples.

**Figure 3 microorganisms-09-00459-f003:**
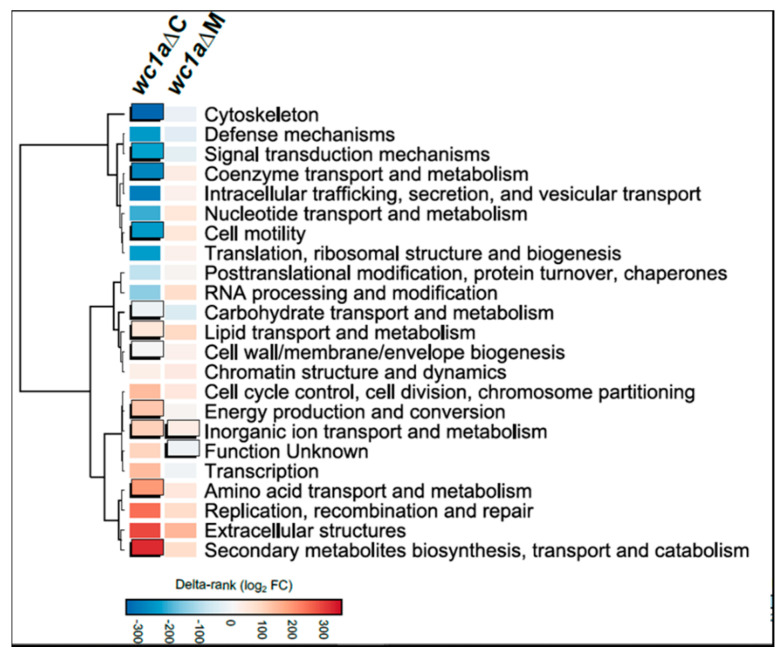
Significant genes during regular growth (*wc1a*Δc) and macrophage phagocytosis (*wc1a*ΔM) were analyzed for Eukaryotic Orthologous Groups (KOG) class enrichment. Significant enrichments (Fisher’s exact test, *p* ≤ 0.05) are shown as uplifted rectangles. A measure of up- or down-regulation for each KOG class is indicated according to a colored scale of delta-rank values (the difference between the mean rank differential expression value of all genes in a particular KOG class and the mean rank differential expression value of all other genes). KOG classes are clustered according to the similarity of their delta rank values.

**Figure 4 microorganisms-09-00459-f004:**
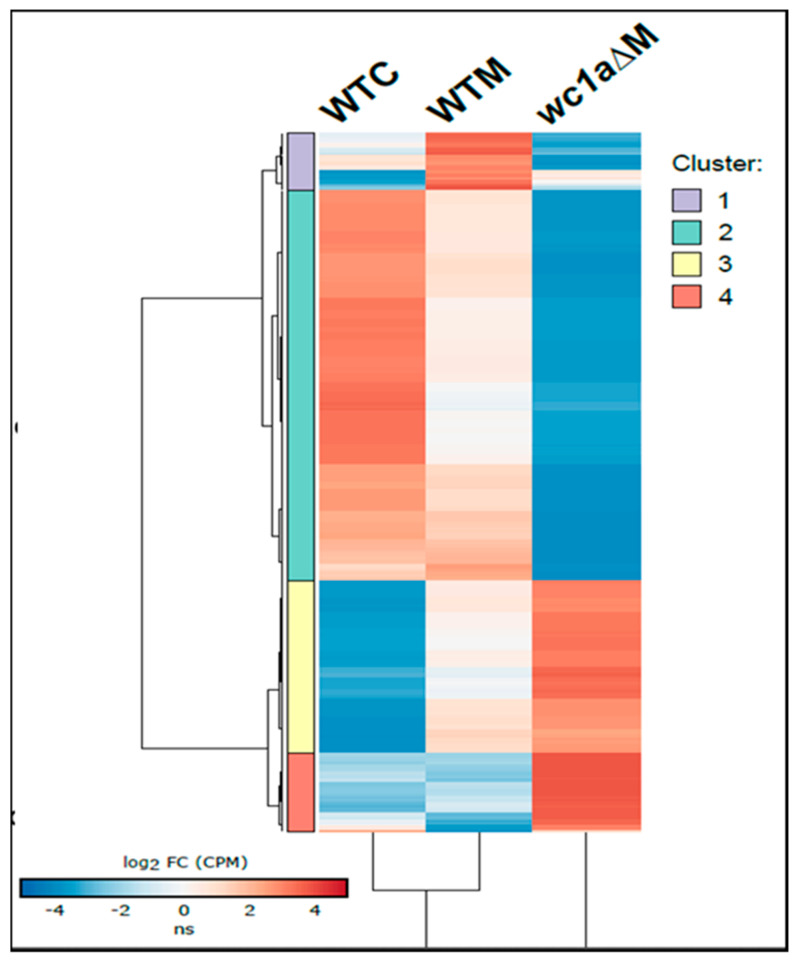
The expression values (calculated as the Z-score of CPM) of significant genes during macrophage phagocytosis are clustered by similarity to analyze phagocytosis response in the *mcwc-1a*Δ mutant. Expression values for wild-type strain in control conditions (WTC) and during macrophage phagocytosis (WTM), as well as for the *mcwc-1a*Δ during macrophage phagocytosis (*wc1a*ΔM), are shown. Four major clusters were identified and shown as colored blocks that indicate the full length of the clusters.

**Figure 5 microorganisms-09-00459-f005:**
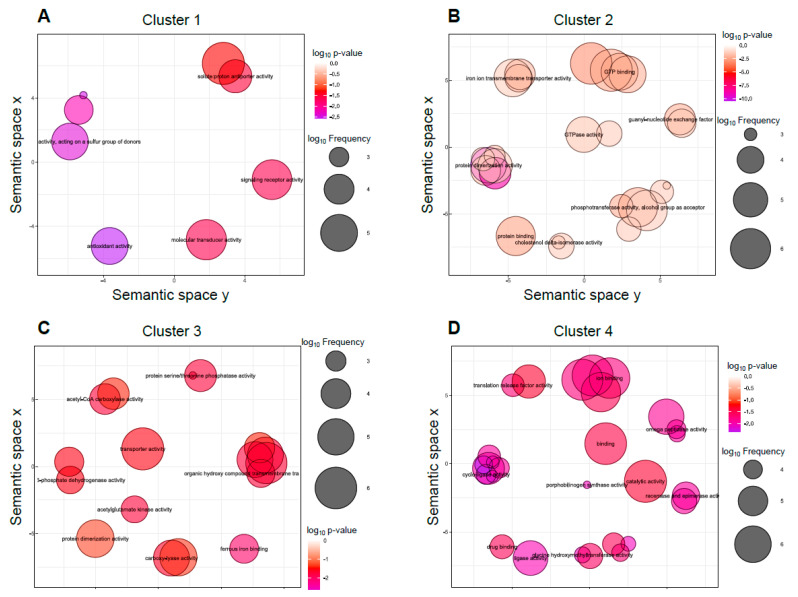
Gene Ontology molecular function terms enrichment in the *mcwc*-1a-dependent response to macrophage phagocytosis. (**A**–**D**) Each dot in the scatter plots represents a molecular function GO term, color-coded to indicate the *p*-value of the enrichment and size-coded to show the frequency of the term in the whole gene set, both in log_10_ units. The plots show GO terms remaining after redundancy reduction. The axes display a bidimensional space that shows semantic similarity among GO terms, i.e., GO terms close together are involved in similar GO molecular functions. Each plot shows GO molecular function enrichment in the genes comprised in clusters 1 (**A**), 2 (**B**), 3 (**C**), and 4 (**D**).

**Table 1 microorganisms-09-00459-t001:** Differentially expressed genes in the *mcwc*-1aΔ mutant strain compared to the wild-type strain.

Culture Conditions	Upregulated Genes ^1^		Downregulated Genes ^2^	
		Average log_2_ FC ^3^		Average log_2_ FC ^3^
L15 5 h 37 °C	1957	2.53 ± 1.48	1499	−1.84 ± 0.86
L15 5 h 37 °C + Mφ	172	1.71 ± 1.10	306	−1.62 ± 0.61

^1^ False discovery rate (FDR) ≤ 0.05, log2 fold change (log_2_ FC) ≥ 1.0, average log_2_ count per million reads (CPM) ≥ 0.0; ^2^ FDR ≤ 0.05, log_2_ FC ≥ −1.0, average log_2_ CPM ≥ 0.0; ^3^ average of all log_2_ fold change values and standard deviation.

## Data Availability

Raw data files generated by this work are deposited at the National Center for Biotechnology Information Sequence Read Archive (NCBI SRA) and are publicly available through the project accession number PRJNA674566. These data were compared to a wild-type strain in the same conditions, previously available at GEO [12] under the following sample accession numbers: GSM3293661 and GSM3293662 (wild-type strain single-cultured); and GSM3293663 and GSM3293664 (wild-type strain co-cultured with mouse macrophages). The Mucci2 [33] genome and annotation files can be accessed at the Joint Genome Institute (JGI) website (http://genome.jgi.doe.gov/ (accessed on 19 February 2021)) and used under the JGI Data Usage Policy.

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
