# Peer review of "A Mucoralean White Collar-1 Photoreceptor Controls Virulence by Regulating an Intricate Gene Network during Host Interactions"

_microorganisms, 2021, doi:10.3390/microorganisms9020459_

Round 1

Reviewer 1 Report

The manuscript explores the role of the light sensing White Collar 1 proteins in the ability of the human pathogenic fungus Mucor lusitanicus to cause disease.  Work in other fungal species had implicated homologs as required for disease, but the situation in M. lusitanicus was more complex because the species has three copies.  The authors found that one of the three genes plays a role in disease development in mice, as deletion of the gene caused a reduction in disease severity.  They then explore what genes may be regulated by this mcwc-1a gene, finding nearly 3,500 genes under its control.  One curious observation from the transcriptomics study was that there was a larger impact of the gene during saprotrophic growth rather than when within macrophages.  The research contributes to a better understanding of how the White Collar 1 proteins impact fungal biology, and will be of interest to those studying pathogenic fungi and light-sensing in fungi.

A main limitation to the research relates to the depth of analysis of the change in virulence in the mutants.

  • The animal experiment was terminated at day 20, prior to the final outcome of the infection with the mcwc-1a mutant could be assessed, i.e. would all of the animals have succumbed to disease if left longer?
  • What is the current requirement in the research community for controls like complemented strains? The experiment would have been stronger if reintroduction of a wild type copy of the mcwc-1a gene restored full virulence.
  • Without creating of multiple gene mutations in a single strain, it is not possible to exclude roles for the other two mcwc-1 genes, as there may be functional redundancy (see comment on line 210 ‘only the gene mcwc-1a has a specialized role in virulence’).

Some minor editorial suggestions are as follows.

Line 13: delete ‘mucomycosis’.

Line 15: delete ‘against mucormycosis’.

Line 17: clearer as ‘Study of the three genes of this family’.

Line 33: delete ‘essentially’.

Line 68 and 198: the word ‘recently’ is ambiguous.  The first report of a link to virulence dates to 2005, so 15 years ago, which does not seem recent (i.e. PMID 15760278). 

Line 81: see also plant pathogenic fungus example Cercospora zeae-maydis (PMID 21829344).

Line 129: add the missing book citation.

Line 244: ‘were’ for ‘was’.

Line 290: in addition to reference 51, also the iron link to virulence in Rhizopus oryzae, e.g. especially FTR1 homolog in PMID 20545847, also may be also PMID 25974051.

Line 308: ‘stands out’ rather than ‘outstands’.

Line 356: ‘completed’ is an exaggeration, as other early lineages in the fungi have WC-1 homologs and are pathogens.

Lines 376-378: the discussion on the zinc finger is confusing: (i) the logic seems to be that the zinc finger is needed for virulence, yet mcwc-1b is not involved in virulence because it does not have a zinc finger, and (ii) MCWC-1B has a zinc finger, but it is non-canonical.  This text may be clearer with some editing.

Line 381: remove italics from ‘and’.

Line 386: reference 23 should be 53.

Line 415: is there therefore an oxidative stress phenotype associated with mutation of the gene?  This would be an easy thing to test and therefore account for the reduction in virulence.

In places the species names are not written using italics font.

Author Response

REVIEWER 1

First of all, thank you very much for sending us comments on the manuscript entitled “A Mucoralean White Collar-1 Photoreceptor Controls Virulence by Regulating an Intricate Gene-Network during Host Interactions” (Microorganisms-1077918).  We want to acknowledge the excellent reviewing effort made with this manuscript. The appreciated list of recommendations helped us generate an improved version of this manuscript.

Comment 1

“The manuscript explores the role of the light sensing White Collar 1 proteins in the ability of the human pathogenic fungus Mucor lusitanicus to cause disease.  Work in other fungal species had implicated homologs as required for disease, but the situation in M. lusitanicus was more complex because the species has three copies.  The authors found that one of the three genes plays a role in disease development in mice, as deletion of the gene caused a reduction in disease severity.  They then explore what genes may be regulated by this mcwc-1a gene, finding nearly 3,500 genes under its control.  One curious observation from the transcriptomics study was that there was a larger impact of the gene during saprotrophic growth rather than when within macrophages.  The research contributes to a better understanding of how the White Collar 1 proteins impact fungal biology, and will be of interest to those studying pathogenic fungi and light-sensing in fungi”.

Response 1:

We really appreciate the time spend by the reviewer for a thorough understanding of the results in our manuscript and for the kind words on the interest aroused from those results.

Comment 2:

“A main limitation to the research relates to the depth of analysis of the change in virulence in the mutants.

  • The animal experiment was terminated at day 20, prior to the final outcome of the infection with the mcwc-1a mutant could be assessed, i.e., would all of the animals have succumbed to disease if left longer?”

Response 2:

The first time that M. lusitanicus was tested in mice-virulence assays was in the study of Charles H. Li et al., Plos Pathogens 2011, in which survival was monitored during 10 days. Later studies increased this period to 20 days. However, it is considered risky to continue the experiments above 20 days as the repetitive immunosuppressions might lead to deaths due to other infections.

Comment 3:

“•        What is the current requirement in the research community for controls like complemented strains? The experiment would have been stronger if reintroduction of a wild type copy of the mcwc-1a gene restored full virulence.”

Response 3:

There are currently two options accepted by the research community regarding the questions raised above: the phenotype can be restored by complementation, or the virulence assay must be done twice with independent mutants. We followed the second option. In fact, the authors M. Sanchis and J. Capilla, from the University of rovira i virgili (Catalonia, Spain), are part of this work because the first infections with the first mutant in mcwc-1a (MU242) were studied by them in their lab. The second infections with the second independent mutant (MU243) were done in our laboratory (University of Murcia, Spain), confirming the same results obtained in their laboratory. However, we failed to describe this information adequately in the manuscript, which has now been corrected and included in the Methods and Materials section.

Comment 4:

“•        Without creating of multiple gene mutations in a single strain, it is not possible to exclude roles for the other two mcwc-1 genes, as there may be functional redundancy (see comment on line 210 ‘only the gene mcwc-1a has a specialized role in virulence’)”.

Response 4:

We agree with this assessment, line 210 has been corrected to: “the gene mcwc-1a has a specialized role in virulence, although a partial redundancy of the other two genes cannot be excluded”.

Other minor comments:

Line 13: delete ‘mucomycosis’

Done.

Line 15: delete ‘against mucormycosis’.

Done.

Line 17: clearer as ‘Study of the three genes of this family’.

Changed.

Line 33: delete ‘essentially’.

Done.

Line 68 and 198: the word ‘recently’ is ambiguous.  The first report of a link to virulence dates to 2005, so 15 years ago, which does not seem recent (i.e. PMID 15760278).

Both have been deleted

Line 81: see also plant pathogenic fungus example Cercospora zeae-maydis (PMID 21829344).

Text and references have been added.

Line 129: add the missing book citation.

The missing reference has been added.

Line 244: ‘were’ for ‘was’.

Done

Line 290: in addition to reference 51, also the iron link to virulence in Rhizopus oryzae, e.g. especially FTR1 homolog in PMID 20545847, also may be also PMID 25974051.

Both references have been added

Line 308: ‘stands out’ rather than ‘outstands’.

Done.

Line 356: ‘completed’ is an exaggeration, as other early lineages in the fungi have WC-1 homologs and are pathogens.

The sentence has been rephrased as: Our results added the analysis of the wc-1 genes to the evolutionary tree

Lines 376-378: the discussion on the zinc finger is confusing: (i) the logic seems to be that the zinc finger is needed for virulence, yet mcwc-1b is not involved in virulence because it does not have a zinc finger, and (ii) MCWC-1B has a zinc finger, but it is non-canonical.  This text may be clearer with some editing.

The text has been clarified: Two independent studies demonstrated in two different fungal models, C. neoformans and Fusarium asiaticum, that the photoreception domains are not required in virulence, whereas the DNA binding zinc-finger domains are essential for full virulence. These results could explain why the mcwc-1b gene of M. lusitanicus is not essential in virulence because this gene encodes for a non-canonical zinc-finger domain that might be involved only in the regulation of carotenogenesis

Line 381: remove italics from ‘and’.

Done.

Line 386: reference 23 should be 53.

All the references have been checked and corrected. It seems that when the template of the journal was used, the references of the original manuscript were misplaced and even deleted.

Line 415: is there therefore an oxidative stress phenotype associated with mutation of the gene?  This would be an easy thing to test and therefore account for the reduction in virulence.

Indeed, we rapidly tested the mutant and wild type in plates with different concentrations of H2O2, but no differences were found. The in vivo scenario is probably more complicated than these rapid tests made in plates, but we will have in mind this suggestion for future experiments.

In places the species names are not written using italics font.

Yes, many have been corrected. Again, when we filled the templated of the journal, the format of the entire paragraphs was automatically changed, and we failed to identify these changed.

Reviewer 2 Report

This data is interesting but very superficial not giving any actual concrete future leads. Highlighting GO terms enrichment is not sufficient to provide interesting data. However, it could lead to interesting finding after in depth analysis, proposition of actual more specific mechanisms involving putative genes, with simple experiments that could support them. Overall, this manuscript should be rewritten, separating more clearly results and discussion (only stating facts and results in the result section) and reducing considerably the introduction to necessary facts only. Here are some additional comments that may help improve this study.

Major comments :

  • Introduction
    • Introduction should be reduced by half and more concise and direct to the point.
  • Methods
    • There is no reference to the previously generated mutant strains. How were they made ?
    • Were white collar-1 genes deleted in R7B, please specify, otherwise it is does not seem correct to have used R7B as wild-type control
    • Host-pathogen in vitro assay : there is no macrophage only control
  • Results
    • Why is there such a big difference in number of gene up and down-regulated between spores alone and spores/macrophages ? What is the total number of identified genes in each group and is it similar ? What is the percentage of gene up and down regulated ? Important decrease of gene up and down regulated between both condition may be artificial if extraction was not as efficient when macrophages are also present.
    • Figure 4 needs to include wc1a∆ control
    • Figure 5 is poorly readable. Figure C and D are also entitled cluster 1 and 2 whereas legend refers to cluster 3 and 4. If these refers to clusters from Figure 4 they need to be named similarly in Figure 4.
    • Please remove references in results section and delete “discussion-like” sentences (i.e. line 195-200, aims and hypothesis are to be stated in the introduction only, line 247-275 is discussion and not results…)
    • More details are needed in 3.1 section : number of mice, % of survival, etc…
    • Supplementary data tables need to be made understandable to the reader (legend)
    • Specific genes need to be highlighted and discussed may be the highest down or up-regulated genes.

Minor comments:

  1. Line 129 : what is “REF book” ?
  2. Line 131 : please specify duration of culture before conidial harvest
  3. There are 2 sections 2.3 in the methods
  4. Figures and Table should not be gathered in a subsection it makes it difficult to read

Author Response

REVIEWER 2

First of all, thank you very much for sending us comments on the manuscript entitled “A Mucoralean White Collar-1 Photoreceptor Controls Virulence by Regulating an Intricate Gene-Network during Host Interactions” (Microorganisms-1077918).  We want to acknowledge the excellent reviewing effort made with this manuscript. The appreciated list of recommendations helped us generate an improved version of this manuscript.

Major comments

Comment 1:

Introduction

Introduction should be reduced by half and more concise and direct to the point.

Response 1:

As suggested, the introduction has been substantially reduced.

Comment 2:

Methods

There is no reference to the previously generated mutant strains. How were they made ?

Response 2:

Clarifications and references have been included: “M. lusitanicus strain CBS277.49 served as the parental strain of all the following strains. The mutants in white collar-1 genes [mcwc-1a∆ (MU242), mcwc-1b∆ (MU244), and mcwc-1c∆ (MU247)] were generated in a previous study [24], derived from the leucine and uracil auxotrophic strain MU402 by rescuing the uracil auxotrophy. MU402 was derived by mutagenesis from the leucine auxotrophic strain R7B, which was used as the wild-type control strain for RNA-seq analyses and virulence assays [25] because it harbors the same genetic background and leucine auxotrophy as the wc-1 mutants, except for the aforementioned deletions. M. lusitanicus strain NRRL3631 is an independent natural isolate that was used as an avirulent mock control in the virulence assays [26].”

Comment 3:

Were white collar-1 genes deleted in R7B, please specify, otherwise it is does not seem correct to have used R7B as wild-type control

Response 3:

These mutants were done in a previous study made in our group and cited in this study. It also has been clarified in the last comment. Briefly, mutants in wc-1 genes were done in MU402, a strain with a double auxotrophy (uracil and leucine) derived from R7B (only auxotrophic for leucine). Thus, when the deletions in MU402 are done with a PyrG marker, the resultant mutants (the three wc-1 genes) are equivalent to R7B, in other words, they are all leucine auxotrophic strains, and they all derive from CBS277.49.

Comment 4:

Host-pathogen in vitro assay : there is no macrophage only control

Response 4:

In this study, we were not investigating the gene response of macrophages after the phagocytosis of mucoralean spore. We studied this response in a previous work entitled “Mucor circinelloides Thrives inside the Phagosome through an Atf-Mediated Germination Pathway”; Pérez-Arques et al., 2019, mBio.

Comment 5:

Results

  • Why is there such a big difference in number of gene up and down-regulated between spores alone and spores/macrophages ? What is the total number of identified genes in each group and is it similar ? What is the percentage of gene up and down regulated ? Important decrease of gene up and down regulated between both condition may be artificial if extraction was not as efficient when macrophages are also present.

Response 5:

We have observed this decrease in differentially expressed genes (DEGs) in M. circinelloides spores confronting phagocytosis in our previous transcriptomic analyses of (Pérez-Arques et al., mBio 2019; Pérez –Arqueset al., Plos Genetics 2020; cited in the manuscript). We reason that the hostile phagosomal environment triggers a very specific stress transduction pathway, resulting in quite similar transcriptomic profiles even when comparing mutants in essential transcription factors and the wild-type strain. On the contrary, when the spores are growing in a non-stressful saprotrophic environment, i.e., in a rich cell culture medium without the stress of the macrophages, the differences between mutant and wild-type strain are more prominent and result in more DEGs. The genes responding to phagocytosis in M. lusitanicus wild-type strain are already discussed in our previous study (Pérez-Arques et al., mBio 2019), but the information regarding DEGs in the mcwc1a mutant in saprotrophic conditions (without macrophages) and during phagocytosis read as follows: “… a total of 3456 genes were directly or indirectly regulated by mcwc-1aΔ (1957 upregulated and 1499 downregulated) during saprotrophic conditions, whereas after phagocytosis, only 478 genes showed differential expression controlled by this gene (172 upregulated and 306 downregulated)”. We have added to the text the following sentence: “There are 11,719 genes annotated, which means that these number of genes correspond to a 29% during saprotrophic conditions (17% upregulated and 12% downregulated) and 4% (1% upregulated and 3% downregulated)”. We agree with the reviewer and thought carefully about that during our transcriptomic analysis. To ensure that all the RNA samples represented a similar proportion of the transcriptional landscape contained in the spore, we normalized the raw reads by the TMM method and analyzed the expression of well-known housekeeping genes (EF-1, TFIIIC, and V-ATPase), which is showing a similar level of expression among samples (yellow dots on the scatterplots). Thus, and adding the robustness of the Bayesian statistical analysis to infer DEGs, we are certain that these differences are not artefactual but real.

Comment 6:

  • Figure 4 needs to include wc1a∆ control

Response 6:

The main focus of this section is to identify transcriptional differences that could explain the reduced-virulence phenotype of the mcwc-1a mutant. To assess this aim, we analyzed the transcriptomic differences between mutant and wild-type strains during macrophage phagocytosis, reckoning it represents a close in vitro environment to the actual in vivo infective process. We concur that the differences in gene expression during saprotrophic growth can be interesting for other researchers, and we have included those data in the supplementary data (Supplementary Table S1). However, to ease the understanding and offer a clear interpretation of our data, we decided to exclude them from our primary figures.

Comment 7:

  • Figure 5 is poorly readable. Figure C and D are also entitled cluster 1 and 2 whereas legend refers to cluster 3 and 4. If these refers to clusters from Figure 4 they need to be named similarly in Figure 4.

Response 7:

We agree Figure 5 needed clarification. It has been entirely rearranged to make it more readable, as suggested. Indeed, cluster labels were incorrect, which has been corrected in accordance with the legend: clusters 1 (A), 2 (B), 3 (C), and 4 (D).

Comment 8:

  • Please remove references in results section and delete “discussion-like” sentences (i.e. line 195-200, aims and hypothesis are to be stated in the introduction only, line 247-275 is discussion and not results…)

Response 8:

Most references, sentences discussion-like, and introduction-like have been deleted as suggested. Exceptions have been made with new references and sentences suggested by other reviewers.

Comment 9:

  • More details are needed in 3.1 section : number of mice, % of survival, etc…

Response 9:

The survival percentages have been added to the text as follows: “In contrast, lacking any of the other two wc-1 genes, mcwc-1b or mcwc-1c, did not alter the virulence potential of M. lusitanicus because the corresponding mutants showed a similar pathogenic behavior as the virulent wild-type R7B strain (survival percentages: mcwc-1aΔ 75%, mcwc-1bΔ 25%, mcwc-1cΔ 25%, R7B 12,5% and NRRL3631 100%)”.

The number of mice and other details such as strain, weight, age, etc., is described in the method section, read as follows: Eight OF-1 mice weighing 30g and two months old (Charles River, Barcelona, Spain) were used as animal models for virulence assays. The mice were immunosuppressed by intraperitoneal administration of cyclophosphamide (200 mg/kg of body weight), 2 days prior to infection and then once every 5 days. Ten-mouse groups were challenged intravenously by retroorbital injection of 1x106 spores”.

Comment 10:

  • Supplementary data tables need to be made understandable to the reader (legend)

Response 10:

The reviewer is absolutely right, and we appreciate the opportunity to correct this oversight on our part. The supplementary data tables now contain a header and a brief legend, as follows:

Table S1. Transcriptomic analysis of M. circinelloides wc-1a deletion mutant and wild-type strain during saprotrophic growth and macrophage phagocytosis. Each gene analyzed is represented by its gene ID (geneId) corresponding to the Mucci2 genome of M. circinelloides available at the Joint Genome Institute website (https://mycocosm.jgi.doe.gov/Mucci2/Mucci2.info.html). The differential expression was calculated as the log2 fold change ratio for each gene in the following comparisons: wildtype strain during macrophage phagocytosis / wildtype strain during saprotrophic conditions (log2FC_R7BM-R7Bc), mcwc-1a mutant strain during macrophage phagocytosis / mcwc-1a mutant strain during saprotrophic growth (log2FC_WC1aM-WC1ac), mcwc-1a mutant strain / wildtype strain during macrophage phagocytosis (log2FC_WC1aM-R7BM), and mcwc-1a mutant strain / wildtype strain during saprotrophic growth (log2FC_WC1aC-R7BC). The p-value and adjuster p-value in a False Discovery Rate test (adj_pvalue) is shown next to each log2 FC comparison. The individual expression values measured as counts per million (CPM) in log2 units are shown for each individual replicate as follows: wild-type strain during saprotrophic growth (duplicates R7Bcb and R7Bca), wild-type strain during macrophage phagocytosis (duplicates R7BMb and R7BMa), mcwc-1a mutant strain during saprotrophic growth (WC1aCb and WC1aCa), and mcwc-1a mutant strain during macrophage phagocytosis (WC1aMb and WC1aMa).

Table S2. Functional annotation of differentially expressed genes in Cluster 1. Each gene analyzed is represented by its gene ID (GeneID) corresponding to the Mucci2 genome, followed by the log2 fold change ratio between the expression in the mcwc-1a mutant and the wild-type strain during macrophage phagocytosis and the adjusted p-value of the statistical False Discovery Rate test. Next, there are functional annotations regarding euKaryotic Orthologous Groups (KOG) class, KOG definition, GO Terms, Interpro protein domains, Enzyme Commission number, Kyoto Encyclopedia of Genes and Genomes (KEGG) orthology (ko) number, KEGG pathway , KEGG module, KEGG reaction, KEGG class, KEGG BRITE functional hierarchy, KEGG Transporter Classification (TC), Carbohydrate-Active enZymes (CAZy), Biochemical Genetic and Genomic (BiGG) reaction, and the probability of harboring a signal peptide (most probable = 1).

Table S3. Functional annotation of differentially expressed genes in Cluster 2. Each gene analyzed is represented by its gene ID (GeneID) corresponding to the Mucci2 genome, followed by the log2 fold change ratio between the expression in the mcwc-1a mutant and the wild-type strain during macrophage phagocytosis and the adjusted p-value of the statistical False Discovery Rate test. Next, there are functional annotations regarding euKaryotic Orthologous Groups (KOG) class, KOG definition, GO Terms, Interpro protein domains, Enzyme Commission number, Kyoto Encyclopedia of Genes and Genomes (KEGG) orthology (ko) number, KEGG pathway , KEGG module, KEGG reaction, KEGG class, KEGG BRITE functional hierarchy, KEGG Transporter Classification (TC), Carbohydrate-Active enZymes (CAZy), Biochemical Genetic and Genomic (BiGG) reaction, and the probability of harboring a signal peptide (most probable = 1).

Table S4. Functional annotation of differentially expressed genes in Cluster 3. Each gene analyzed is represented by its gene ID (GeneID) corresponding to the Mucci2 genome, followed by the log2 fold change ratio between the expression in the mcwc-1a mutant and the wild-type strain during macrophage phagocytosis and the adjusted p-value of the statistical False Discovery Rate test. Next, there are functional annotations regarding euKaryotic Orthologous Groups (KOG) class, KOG definition, GO Terms, Interpro protein domains, Enzyme Commission number, Kyoto Encyclopedia of Genes and Genomes (KEGG) orthology (ko) number, KEGG pathway , KEGG module, KEGG reaction, KEGG class, KEGG BRITE functional hierarchy, KEGG Transporter Classification (TC), Carbohydrate-Active enZymes (CAZy), Biochemical Genetic and Genomic (BiGG) reaction, and the probability of harboring a signal peptide (most probable = 1).

Table S5. Functional annotation of differentially expressed genes in Cluster 4. Each gene analyzed is represented by its gene ID (GeneID) corresponding to the Mucci2 genome, followed by the log2 fold change ratio between the expression in the mcwc-1a mutant and the wild-type strain during macrophage phagocytosis and the adjusted p-value of the statistical False Discovery Rate test. Next, there are functional annotations regarding euKaryotic Orthologous Groups (KOG) class, KOG definition, GO Terms, Interpro protein domains, Enzyme Commission number, Kyoto Encyclopedia of Genes and Genomes (KEGG) orthology (ko) number, KEGG pathway , KEGG module, KEGG reaction, KEGG class, KEGG BRITE functional hierarchy, KEGG Transporter Classification (TC), Carbohydrate-Active enZymes (CAZy), Biochemical Genetic and Genomic (BiGG) reaction, and the probability of harboring a signal peptide (most probable = 1).

Comment 11:

  • Specific genes need to be highlighted and discussed may be the highest down or up-regulated genes.

Response 11:

We agree that a gene-by-gene dissection and functional analysis of our transcriptomic data holds the key to completely characterize the role of mcwc-1a in regulating virulence during macrophage phagocytosis, and probably in other molecular processes. That is our aim for future endeavors, but we anticipate it will be extremely resource and time-consuming. As a more straightforward alternative, we decided to analyze the data as a whole and study the functional categories and GO terms that are enriched in the response of the mcwc-1a mutant to phagocytosis, providing the basis for future research. Functional enrichment is a frequent technique employed as downstream analyses on collections of genes identified by high-throughput transcriptomic methods. However, to facilitate a gene-by-gene search and discussion, we have included several supplementary data tables (Supplementary Tables S2-S5) listing all the genes contained in each cluster identified in Figure 4, as well as a descriptive functional annotation.

Minor comments:

  1. Line 129 : what is “REF book” ?

Correct reference has been added

  1. Line 131 : please specify duration of culture before conidial harvest

Data has been included as follows: “Spores were harvested after six days and then filtered using Falcon® 70 µm cell strainers to remove unintentionally harvested mycelium pieces before confronting macrophages or animal models.”

  1. There are 2 sections 2.3 in the methods

Corrected

  1. Figures and Table should not be gathered in a subsection it makes it difficult to read

This is a mandatory format in the template provided by the journal and recommended in the author's instructions.

Reviewer 3 Report

The manuscript by Pérez-Arques et al. describes the transcriptional response of Mucor lusitanicus dependent on mcwc-1a during phagocytosis by macrophages. Authors claim that mcwc-1a regulates transcription of several genes involved in virulence, namely genes involved in oxidative response. Mutant mcwc-1a is much less virulent than wild type or mutants for the other two mcwc-1 paralogs, suggesting a specialized role for mcwc-1a in virulence. The results depicted in here are the first step in the identification of new drug targets that could serve for the development of new antifungal compounds.

The paper is well written with some minor errors and the results depicted in figures are well presented and discussed. The conclusions are supported by the presented results.

Major point:

Authors should mention how does the expression of wc-1 genes vary in the different wc-1 mutants? Are mcwc-1b and mcwc-1c genes differently expressed upon mcwc-1a deletion?  Is there any compensation between the different paralogs?

Do mutant spores germinate inside the phagocytic cells?

Minor points:

Line 129, sporulation (REF book).

Line 146, wild type and wc-1a mutant?

Line 191, references are missing

Line 234, “regulated by mcwc-1aΔ” should read, “regulated by mcwc-1a

Line 244, “Once the genes regulated by mcwc-1aΔ was” should read, “Once the genes regulated by mcwc-1a was”

Line 273, “them into 4 clusters (Tables S1, S2, S3, and S4)” should read, “them into 4 clusters (Tables S2, S3, S4, and S5)”

Figure 5, only 2 of the 4 clusters are indicated in the figure although figure legend says that all 4 clusters are represented. Please change accordingly.

Supplementary tables should have titles describing which data is depicted in there. Since authors refer cluster 1 composed by 39 genes in the first place maybe these results should be depicted in table S2. Then, cluster 2 genes will compose table S3, cluster 3 table S4 and cluster 4 table S5.

Author Response

REVIEWER 3

First of all, thank you very much for sending us comments on the manuscript entitled “A Mucoralean White Collar-1 Photoreceptor Controls Virulence by Regulating an Intricate Gene-Network during Host Interactions” (Microorganisms-1077918).  We want to acknowledge the excellent reviewing effort made with this manuscript. The appreciated list of recommendations helped us generate an improved version of this manuscript.

Comment 1

“The manuscript by Pérez-Arques et al. describes the transcriptional response of Mucor lusitanicus dependent on mcwc-1a during phagocytosis by macrophages. Authors claim that mcwc-1a regulates transcription of several genes involved in virulence, namely genes involved in oxidative response. Mutant mcwc-1a is much less virulent than wild type or mutants for the other two mcwc-1 paralogs, suggesting a specialized role for mcwc-1a in virulence. The results depicted in here are the first step in the identification of new drug targets that could serve for the development of new antifungal compounds.

The paper is well written with some minor errors and the results depicted in figures are well presented and discussed. The conclusions are supported by the presented results”.

Response 1:

We really appreciate the time spend by the reviewer for a thorough understanding of the results in our manuscript and for the kind words on the interest aroused from those results.

Comment 2:

Authors should mention how does the expression of wc-1 genes vary in the different wc-1 mutants? Are mcwc-1b and mcwc-1c genes differently expressed upon mcwc-1a deletion?  Is there any compensation between the different paralogs?

Response 2:

The expression of the three paralogs of mcwc-1 gene family in the WT and the three mutants was thoroughly dissected in the previous study of Silva et al., 2006, Molecular Microbiology (can be found in the reference list of this work). They found that  mcwc-1c gene expression was severely affected in the mcwc-1a null mutant only in the light. Thus, in the mcwc-1a mutant,  the mcwc-1c mRNA levels in the light-induced mycelia were much lower than in the wild-type and mcwc-1b null mutant strains. This result clearly indicated that the mcwc-1a gene is involved in the regulation of mcwc-1c expression by light, but not in the dark. In this work, both the in vivo assays (infections in mice hosts) and the in vitro macrophages-spores interactions in cell culture were done in dark conditions, ruling out any compensation effects in the expression of the three paralogs.

Response 2:

Do mutant spores germinate inside the phagocytic cells?

Comment 2:

In a previous study (Trung et al., 2017 Plos Pathogens;  also can be found in the references list), we developed an assay to measure spore germination inside the phagocytic cells, finding that many genes involved in virulence presented delayed germination. Thus, one of the first experiments that we did in this work was to measure spore gemination of the three mutants inside the phagocytic cells, finding that all of them were similar to the wild type. These results mean that the regulation of virulence by mcwc-1a must be found downstream of the germination, for instance, controlling the tropism during tissue invasion as we suggest in the discussion.

Minor points:

Line 129, sporulation (REF book).

Correct reference has been added to the text

Line 146, wild type and wc-1a mutant?

Yes, wild type has been added to the sentence.

Line 191, references are missing

References have been added

Line 234, “regulated by mcwc-1aΔ” should read, “regulated by mcwc-1a”

Corrected

Line 244, “Once the genes regulated by mcwc-1aΔ was” should read, “Once the genes regulated by mcwc-1a was”

Corrected

Line 273, “them into 4 clusters (Tables S1, S2, S3, and S4)” should read, “them into 4 clusters (Tables S2, S3, S4, and S5)”

Corrected

Round 2

Reviewer 2 Report

Thanks for the improvements on superficial matters and for clarifying details I probably did not understand at first sight.